# Platelet *Versus* Megakaryocyte: Who Is the Real Bandleader of Thromboinflammation in Sepsis?

**DOI:** 10.3390/cells11091507

**Published:** 2022-04-30

**Authors:** Cédric Garcia, Baptiste Compagnon, Michaël Poëtte, Marie-Pierre Gratacap, François-Xavier Lapébie, Sophie Voisin, Vincent Minville, Bernard Payrastre, Fanny Vardon-Bounes, Agnès Ribes

**Affiliations:** 1Laboratoire d’Hématologie, Centre Hospitalier Universitaire de Toulouse, 31059 Toulouse, France; garcia.cedric@chu-toulouse.fr (C.G.); voisin.s@chu-toulouse.fr (S.V.); bernard.payrastre@inserm.fr (B.P.); 2Institut des Maladies Métaboliques et Cardiovasculaires, Inserm UMR1297 and Université Toulouse 3, 31024 Toulouse, France; baptiste.compagnon@gmail.com (B.C.); michaelpoette@gmail.com (M.P.); marie-pierre.gratacap@inserm.fr (M.-P.G.); bounes.f@chu-toulouse.fr (F.V.-B.); 3Pôle Anesthésie-Réanimation, Centre Hospitalier Universitaire de Toulouse, 31059 Toulouse, France; minville.v@chu-toulouse.fr; 4Service de Médecine Vasculaire, Centre Hospitalier Universitaire de Toulouse, 31059 Toulouse, France; lapebie.fx@chu-toulouse.fr

**Keywords:** platelets, megakaryocytes, thromboinflammation

## Abstract

Platelets are mainly known for their key role in hemostasis and thrombosis. However, studies over the last two decades have shown their strong implication in mechanisms associated with inflammation, thrombosis, and the immune system in various neoplastic, inflammatory, autoimmune, and infectious diseases. During sepsis, platelets amplify the recruitment and activation of innate immune cells at the site of infection and contribute to the elimination of pathogens. In certain conditions, these mechanisms can lead to thromboinflammation resulting in severe organ dysfunction. Here, we discuss the interactions of platelets with leukocytes, neutrophil extracellular traps (NETs), and endothelial cells during sepsis. The intrinsic properties of platelets that generate an inflammatory signal through the NOD-like receptor family, pyrin domain-containing 3 (NLRP3) inflammasome are discussed. As an example of immunothrombosis, the implication of platelets in vaccine-induced immune thrombotic thrombocytopenia is documented. Finally, we discuss the role of megakaryocytes (MKs) in thromboinflammation and their adaptive responses.

## 1. Introduction

While the role of platelets in hemostasis and their activation sequence have been largely investigated, their contribution to the inflammatory process is still the subject of intense research. Many studies have shown their role as immune players in various pathologies, such as atherosclerosis, cancer, stroke, myocardial infarction, and sepsis [1,2,3,4]. The most common types of sepsis are those caused by staphylococci and streptococci and those caused by Gram-negative bacilli such as *Escherichia coli*. The pathophysiology of sepsis is complex and involves several stakeholders, including bacterial or viral components acting as danger signals (pathogen-associated molecular patterns, PAMPS, and damage-associated molecular patterns, DAMPS), the endothelium, and innate immune cells [5,6]. Because platelets are emerging as key players in the immune response of sepsis, as well as in its complications, they hold a position of unquestionable interest.

In this review, we present the ambivalent role of platelets in sepsis with, on the one hand, their interactions with pathogens as a host defense function and, on the other hand, their contribution to the complications of sepsis, such as disseminated intravascular coagulation (DIC), microvascular thrombosis, and organ failure (Table 1). We discuss the reciprocal interactions of platelets with innate immune cells, endothelial cells (ECs), and during the formation of neutrophil extracellular traps (NETs). We also consider the formation of a platelet NOD-like receptor family, pyrin domain-containing 3 (NLRP3) inflammasome and the implication of platelets in an immunothrombosis condition, vaccine-induced immune thrombotic thrombocytopenia. Finally, we speculate on the potential role played by MKs in these thromboinflammation processes.

## 2. Immunothrombosis or Thromboinflammation?

The difference between thromboinflammation and immunothrombosis is quite subtle, and some authors tend to propose to use them indistinguishably [7]. Others justify a distinction [8,9] and propose that immunothrombosis is an intrinsic part of the more general mechanism termed thromboinflammation, as previously described [10,11].

Engelmann and Massberg [9] suggested that immunothrombosis is a major component of the innate intravascular immune system and that it serves different functions: (i) it helps to capture and trap pathogens circulating in the blood, particularly in the microvessels through the fibrin network, thus limiting their spread and tissue invasion; (ii) it generates, through intravascular thrombi, a distinct compartment that promotes pathogen killing using antimicrobial strategies provided by innate immune cells and antimicrobial peptides, generated upon activation of blood coagulation and/or released by activated platelets at sites of pathogen immobilization [12]; (iii) through the microvascular deposition of fibrinogen or fibrin, it promotes the recruitment of additional immune cells to the site of infection and/or tissue injury, enhancing pathogen recognition and coordinating the immune response [13].

Hereafter, we use the term immunothrombosis when the trigger specifically involves the molecular mechanisms of innate immunity. The term thromboinflammation is used when thrombotic processes specifically engage a cytokine storm or any non-infectious inflammatory trigger (chemokine, complement, HMGB1, etc.).

**Table 1 cells-11-01507-t001:** Beneficial and deleterious consequences in platelet-mediated immunothrombosis and thromboinflammation. Some good and bad effects of platelets are secondary to the mechanisms of thromboinflammation and immunothrombosis.

POSITIVE EFFECTS	REFERENCES	NEGATIVE EFFECTS	REFERENCES
Promote leukocyte transmigration by interacting with endothelial cells	[14,15,16,17,18,19]	Neutrophil activation by aggregating with them	[20,21,22,23,24,25,26]
Fill endothelial fenestration with TLT-1 to prevent edema and microhemorrhage	[27,28]	Soluble TLT-1 precipitates plasma fibrinogen	[27,29]
Catch pathogens thanks to their integrins and membrane receptors	[30,31,32,33,34,35,36]	Triggers NETs and acute thrombosis	[37,38,39,40,41,42,43,44,45]
Phagocyte pathogens	[46,47,48,49,50,51]	Procoagulant activity of their membrane Production of microparticles	[52,53,54,55,56]
Production of interleukin (IL)1β and IL18 in response to PAMPs and DAMPs	[19,57,58,59]	Exacerbation of leukocyte inflammasome	[38,60,61]
Tissue repair and regeneration by the local release of the α-granule content	[62,63,64,65,66,67]	Endothelial barrier disruption and organ damage caused by hemorrhage and microthrombosis	[5,7,20,68,69,70,71,72,73,74]

## 3. Stakeholders in the Septic Scene

The septic scene is an inflammatory disorder mediated by the activation of the innate immune system, and its response is characterized by two major elements. First, sepsis is classically initiated by the simultaneous recognition of multiple microbial products and endogenous danger signals by the complement system and by specific cell surface receptors [75]. Innate immune cells, as well as epithelial cells, and endothelial gatekeeper populations are constantly exposed to their local environment. The binding of PAMPs or DAMPs to the complement system, Toll-like receptors (TLRs), nucleotide-binding oligomerization domain (NOD)-like receptors, retinoic acid-inducible gene (RIG)-like receptors, mannose-binding lectin, and/or scavenger receptors induces complex intracellular signaling pathways with overlapping and complementary pathways [76]. Second, the activation of multiple signaling pathways ultimately leads to the expression of the common classes of genes involved in inflammation, adaptive immunity, and/or cellular metabolism. For example, signaling pathways converging at the nuclear translocation of nuclear factor kappa B (NF-κB) are responsible for the early induction of the expression of several genes, including cytokines, associated with their inflammatory responses. These cytokines induce the expression of other inflammatory cytokines and chemokines, as well as the polarization of specific T cells modulating adaptive immunity. The rapid activation of these signaling pathways upon the recognition of PAMPs and DAMPs represents immunothrombosis. The combined activation of the sentinel receptors of the innate immune system, as well as complement cascade activation [77], enhanced by the production of inflammatory cytokines, has a major effect on coagulation and the vascular and lymphatic endothelium, leading to an increased expression of adhesion molecules [78,79]. In parallel, the production of pro-inflammatory proteases leads to the internalization of vascular endothelial (VE)–cadherin, responsible for the loss of tight junctions and vascular permeability [80]. Moreover, immunothrombosis is triggered and maintained by the local accumulation of innate immune cells (particularly monocytes and neutrophils), a process that is likely to involve the adoption of a pro-adhesive phenotype by microvascular ECs that are exposed to pathogens. In this context, platelets act as versatile effectors of antibacterial immunity, and they contribute to bactericidal and bacteriostatic action in synergy with other cells of innate immunity (Figure 1).

**Basal:** ECs secrete nitric oxide (NO) and prostaglandin I2 (PGI2) to inhibit platelet activation [81,82]. They express receptors on their surface to activate anticoagulant factors while blocking coagulant factors. α-thrombin (FIIa), by binding to thrombomodulin (TM) and the endothelial protein C receptor (EPCR), cleaves and activates protein C (PCa) [83,84]. Antithrombin (AT), by binding to EC GAGs, inactivates FIIa and FXa [85]. The tissue factor pathway inhibitor (TFPI) in the endothelium inhibits the tissue factor (TF)/FVIIa complex, blocking the initiation of coagulation [70,71,72,73,86,87]. **Activation:** PAMPs are constitutively expressed on the bacterial membrane, outer membrane vesicles (OMVs), and viral envelope/capsid. DAMPs are produced by damaged cells during infection. Both PAMPs and DAMPs are triggers for the activation of the cellular environment (endothelial cells and platelets). This activation cascade is mediated by TLRs on platelets and ECs [88]. Platelet TLRs enable the activation of the NLRP3 inflammasome for the production of pro-inflammatory cytokines, such as IL1β, as well as the recruitment of GPIb, GPIIbIIIa, and CD40L to the platelet surface for the binding to the endothelium by P-selectin, von Willebrand factor (vWF), and CD40, respectively [89,90]. Damaged ECs can also inactivate their tight junctions, bind IL1β to IL1R, and thus express TF [91], allowing the activation of FVIIa and FXa and then the generation of thrombin. Finally, FIIa may, in turn, promote platelet activation and recruitment [53,54,55]. **Adhesion:** The recruitment of platelets and neutrophils is followed by their adhesion by the complexes P-selectin/PSGL-1, GPIb-IX-V/fibrinogen/MAC-1, and GPIb/MAC-1 [14,15,16,69,92]. This binding generates an activation signal of the neutrophil NLRP3 inflammasome responsible for IL1β and IL18 synthesis [93]. In contrast, cathepsin G produced by activated neutrophils can compromise these interactions by cleaving GPIb or PSGL-1. This cell adhesion allows permeability and endothelial edema, thus promoting the transmigration of platelet and neutrophil cells [78,79,80].

### 3.1. From Leuko–Platelet Interactions to the Involvement in NETosis

One of the most studied and important mechanisms of the platelet response to infection lies in platelet–neutrophil interactions, which induce and/or enhance many of the antibacterial functions of neutrophils [30,31]. These interactions are orchestrated by surface proteins and secreted molecules. Activated platelets express CD62-P on their surface, which specifically binds to the neutrophil membrane protein P-selectin glycoprotein ligand 1 (PSGL-1) [94]. This interaction has been confirmed in septic and acute respiratory failure experimental mice models [69,92]. Similarly, platelet GPIb [95] and junctional adhesion molecule 3 [96] can interact directly with leukocyte membrane αMβ2 or indirectly with integrin αIIbβ3 via fibrinogen as a “bridge” molecule [97]. As a down-regulation mechanism, secreted molecules such as cathepsin G produced by activated neutrophils can compromise these interactions by the cleavage of GPIb or PSGL-1 [14]. Several platelet-derived products control neutrophil recruitment, activation, and function. CD40L secreted by platelets positively regulates neutrophil integrin expression [90]. Serotonin and CXCL4 are involved in neutrophil recruitment by platelets in animal models of inflammation and acute pancreatitis [98]. Moreover, Assinger et al. demonstrated that platelets, activated by the pathogens of periodontitis, increased bacterial clearance by more than 20% via direct interactions with the bacteria, in connection with platelet TRL2 [99]. Gaertner et al. demonstrated for the first time in 2017 that platelets can aggregate bacteria and promote their phagocytosis by neutrophils [100]. Platelets also actively participate in the diapedesis of neutrophils during sepsis. Sreeramkumar et al. demonstrated that neutrophils recruited to inflamed venules have clusters of PSGL-1 on their vascular side that are scanned by circulating activated platelets [92]. Once platelet–neutrophil interactions have effectively occurred, an outside-in signal leads to Mac-1 (αMβ2 integrin) and CXCL2 receptor redistribution, generating polarized receptor microdomains essential for targeting diapedesis to the infected site.

Another antimicrobial mechanism associated with platelet–neutrophil interactions is the formation of neutrophil extracellular traps (NETs). NETs result from the ejection of chromatin loaded with proteolytic enzymes and other antibacterial molecules from the nucleus of activated neutrophils [39,40,101]. A molecular mechanism underlying nucleosome disassembly is histone H3 and H4 citrullination caused by peptidyl arginine deamination 4 (PAD4) activation [41,42]. Moreover, the released neutrophil elastase contributes to chromatin decondensation and histone cleavage. Finally, the disintegration of the nuclear envelope and the rupture of the cytoplasmic membrane release NETs [43,102]. In some situations, this lytic pathway can lead to “cell suicide” caused by NETosis; in other cases, it is a non-lytic pathway considered “vital” [39,40] and able to ensnare and lyse pathogens (bacteria, fungi, parasites, and viruses) [103]. The co-incubation of platelets and neutrophils from healthy donors with septic patient plasma promotes platelet–neutrophil adhesion in a TLR4-dependent way [23,24]. Furthermore, in a model of LPS-induced NETosis, platelets were found to increase the ability of NETs to trap E. coli [23]. Platelet-induced NETosis has been observed in the presence of platelet agonists (thrombin, ADP, arachidonic acid, and collagen), as well as in the presence of TLR ligands. In addition to these direct platelet–neutrophil interactions, TxA2 and RANTES (CCL5) and PF4 (CXCL4) chemokines are also platelet inducers leading to NET formation [23,24,25,26]. Moreover, activated platelets present HMGB1 to neutrophils to engage them in NET generation and autophagy [25]. In a sepsis mouse model, complement C3a was found to be an important actor in platelet activation, while C5a and the procoagulant activity of platelets were found to induce NET expression [44]. NETs also act as a catalytic platform stimulating the proteolytic activity of neutrophil elastase consisting of a procoagulant effect through the degradation of tissue factor pathway inhibitor (TFPI); the resulting factor Xa activation contributes to intravascular fibrin production. This phenomenon is amplified by local thrombin produced from the induction of the contact phase by polyphosphates released by activated platelets [20,104]. The latter is also activated by the binding of histone H3 released by NETs to C-type lectin-like 2 (CLEC-2). All these pathways contribute to the localized formation of thrombi and to trap pathogens.

### 3.2. Platelet and Endothelial Cell (EC) Crosstalk during Sepsis

The preservation of endothelial barrier integrity and blood flow continuity is largely disrupted during sepsis. Besides the loss of endothelial glycocalyx (a heparan sulfate-rich layer of glycosaminoglycans and proteoglycans coating the endothelium) integrity [70,71,72,73,87] and specific anticoagulant factors (antithrombin and TFPI), there is unfavorable deregulation of the fibrinolytic/antifibrinolytic balance, which is further aggravated by leuko–platelet interactions and the generation of NETs [20,21,22]. PAMPs and DAMPs associated with bacterial infection impair the structure and function of the endothelium [88]. Thus, activated ECs release a significant amount of von Willebrand factor (vWF), which controls platelet arrest and clot formation. The high-affinity receptor for thrombin, thrombomodulin, responsible for inhibiting its procoagulant function, is also enzymatically cleaved [52]. Concomitantly, ECs expose P-selectin on their surface, allowing the adhesion of neutrophils through P-selectin–P-selectin glycoprotein ligand-1 (PSGL-1) interactions [14,17,18]. The expressions of E-selectin, intercellular adhesion molecule-1 (ICAM-1), and vascular cell adhesion molecule-1 (VCAM-1) on the luminal endothelial surface allow the adhesion and subsequent transendothelial migration of systemic immune cells (neutrophils, macrophages, and T cells) [15,16]. These infiltrated cells, together with ECs, secrete a plethora of soluble mediators, including cytokines (such as tumor necrosis factor and interleukin-1β), chemokines, growth factors, eicosanoids, proteolytic enzymes, and peroxidases, which influence the inflammatory response. This procoagulant and antifibrinolytic phenotype shift [53,54,55] favors DIC and microvascular thrombosis, which is exacerbated by the deposition of NETs and platelet activation, resulting in a decrease in blood supply to tissues and multiple organ dysfunction [7,20,23,68,69]. Thrombocytopenia resulting from the consumption of platelets combined with the exhaustion of coagulation factors can also lead to severe bleeding. Another aspect of this process is the alteration of activated EC permeability [74]. This hyperpermeability of capillaries results in a massive leakage of plasma fluids into the extravascular space. Vasodilation of the microcirculation alters blood supply and contributes to poor organ perfusion and, ultimately, to a state of septic shock. The supply of intravascular fluids to maintain arterial blood pressure contributes secondarily to the development of edema due to capillary leakage.

In this platelet–endothelial cell relationship, platelets have two proteins in their αgranules that promote the crossing of white blood cells through the endothelium. The expression of TREM-like transcript-1 (TLT-1) on the membrane of activated platelets focuses neutrophil diapedesis to septic lung sites [27]. This protein is believed to complete the P-selectin action in neutrophil transmigration during platelet activation and secretion at the site of infection. Once neutrophil transmigration has taken place, a cleavage occurs and releases neutrophils for further migration to septic tissue areas. For example, an injection of LPS into healthy mice was found to induce an increase in soluble fragment TLT-1 (sTLT-1) in their plasma. The sTLT-1 fragment, released locally from the platelet membrane, precipitated with plasma fibrinogen to form a protein complex covering the areas of diapedesis, limiting microbleeds [93]. The monitoring of plasma sTLT-1 could serve as a marker of the deregulation of these processes in acute respiratory disease syndrome (ARDS) associated with bronchoalveolar hemorrhage. A system of the reuptake of excess soluble TLT-1 by platelets ensures an equilibrium. The saturation of this uptake system in ARDS [29] and sepsis leads to an accumulation of sTLT-1 and the deposition of the fibrinogen/sTLT-1 complex in the bloodstream, outside of the inflammatory sites. This results in disseminated intravascular coagulation [27].

### 3.3. A Platelet Inflammasome during Sepsis?

First described in 2002, the inflammasome functions as an intracytoplasmic multi-protein complex activated by cellular stress [105]. It is an essential component of the innate immune system, allowing it to initiate the inflammatory cascade by the release of cytokines [106]. To date, the NOD-like receptor family, pyrin domain-containing 3 (NLRP3) complex is the best-characterized inflammasome and consists of the assembly of the NLRP3 protein with cysteine protease pro-caspase-1 via the apoptosis-associated speck-like protein containing a CARD (ASC). Its activation leads to the cleavage of pro-caspase-1 into caspase-1 and the transformation of some cytokines into their active forms, including IL1β and IL18 [57]. Its existence in platelets is still debated, but several studies describe the NLRP3 inflammasome as an intrinsic component of platelets, responsible for inflammatory signal propagation.

The platelet NLRP3 inflammasome was described in 2013 in patients infected with dengue virus leading to the synthesis of IL1β through caspase-1 activation. In these conditions, this platelet secretion of IL1β increased endothelial permeability and plasma leakage [107]. More recently, domain 3 of the dengue viral envelope was found to be a virulence factor responsible for thrombocytopenia and the activation of the NLRP3 platelet inflammasome in a mouse model [58]. Thus, during infection with the dengue virus, platelets, through the action of their inflammasome NLRP3 and IL1β synthesis, contribute to the inflammatory signal [58].

The NLRP3 inflammasome has also been described in platelets derived from bacterial sepsis generated either in vitro by lipopolysaccharides (LPS) or in vivo by cæcal ligation and puncture in rats. *In vitro*, platelet activation status was correlated with inflammation and endothelial permeability [19]. LPS, a major component of the outer membrane of Gram-negative bacilli, can generate an in vitro sepsis model and promote platelet activation by Toll-like receptor 4 (TLR4) [108]. The signaling pathway downstream of TLR4 involves various kinases and the MyD88 adaptor molecule to achieve the splicing of IL1β mRNA and, in turn, a rapid release of IL1β [109]. The specific blockage of the TLR4-Syk-NLRP3 pathway in platelets treated with LPS reduces IL1β synthesis [59]. Similarly, the inhibition of NLRP3 in LPS-stimulated septic rat platelets reduces IL1β synthesis and platelet activation, suggesting a correlation between NLRP3 and platelet activation, but the trigger remains to be identified [19]. In septic rats, an antagonist of the adenosine diphosphate (ADP) receptor P2Y12, clopidogrel, has been shown to reduce the activity of the platelet NLRP3 inflammasome, as well as the circulating concentration of IL1β and IL18 and sepsis-induced kidney injury, supporting an immune role of platelets in sepsis through their NLRP3 inflammasome [110]. Interleukin-1β released by platelets can act on other cell types but can also act directly on neighboring platelets. Platelet activation, in response to recombinant LPS-IL1β co-stimulation, results from a dual mechanism comprising a direct aspect, by the activation of platelet IL1β receptors, and an indirect aspect, related to the stimulation of platelet pro-IL1β mRNA splicing capabilities ensuring increased IL1β production. Inhibition of the IL1 receptor or caspase-1 reduces this autocrine loop [109]. In a murine model of thrombosis, NETs, through their histones, could activate platelet caspase-1 through ⍺IIbβ3 integrin [37].

Whether NETs in sepsis promote the activation of the platelet NLRP3 inflammasome is unknown. In a recent study, it was shown that platelets do not have the NLRP3 inflammasome but strongly promote the NLRP3 inflammasome in monocytes [38]. Thus, the real impact of the NLRP3 inflammasome on platelets remains to be further characterized in sepsis.

### 3.4. Effects of Antiplatelet Drugs in Sepsis

Targeting platelet activation in sepsis with conventional inhibitors has been extensively studied. If we focus on the results of meta-analyses published over the last 10 years from cohorts of sepsis patients, aspirin is consistently associated with a reduction in mortality, both on the ward and in the ICU [111,112,113,114]. Regardless of the origin of the sepsis, the effect attributed to aspirin is better than that of other antiplatelet agents [112]. Distinctions between antiplatelet agents, excluding aspirin, have not been extensively studied, and no distinction has been identified, particularly between ticagrelor and clopidogrel [115]. Based on these considerations, and to test the long-term effect of low-dose primary prevention with aspirin, the Aspirin To Inhibit Sepsis Study (ANTISEPSIS) [116] and the wider ASPirin in Reducing Events in the Elderly (ASPREE) [117] study on primary prevention with low-dose aspirin were carried out. The results of the clinical trials are reasonably disappointing, as no beneficial effects were demonstrated in improving sepsis endpoints, and the long-term use of aspirin for the primary prevention of sepsis could have potential side effects such as bleeding [118].

Other antiplatelet agents targeting the ADP receptor P2Y12 may be more effective since they have been shown to reduce deleterious leuko–platelet interactions, as well as the synthesis of inflammatory markers [119], and to improve lung function in a mouse model of lung disease [118]. Recently, in the context of the SARS-CoV-2 epidemic, Omarjee et al. raised the question of the benefit of using ticagrelor to decrease leuko–platelet aggregates, NET release, and capillary leakage to prevent coagulopathy secondary to sepsis in a randomized clinical trial, given the positive effects obtained when administered at the onset of COVID-19 [120].

Advances in identifying the pathways involved in thromboinflammation suggest that ITAM (immunoreceptor tyrosine-based activation motif)-linked receptors are potential therapeutic targets. Thus, CLEC-2 and GPVI could be particularly interesting, with a possible advantage for CLEC-2, as demonstrated in murine models of sepsis (LPS injection or cæcal ligation puncture) where the platelet CLEC-2 receptor is knocked out or inhibited by a monoclonal antibody, with no increased bleeding risk [117].

Finally, given the role of leuko–platelet interactions in the mechanisms associated with thromboinflammation, they could also be relevant targets. Preclinical studies testing P-selectin, GPIb, integrin α2β3, PSGL-1, and Mac-1 show a reduction in leuko–platelet aggregates, as well as an improvement in microcapillary dysfunction and the resolution of some inflammation markers. For example, a phase 1 trial has already confirmed the safety and benefit of inclacumab, a monoclonal antibody to P-selectin [121].

## 4. Vaccine-Induced Immune Thrombotic Thrombocytopenia: An Example of Immunothrombosis Involving Platelets

In response to the SARS-CoV-2 pandemic, several laboratories have developed different kinds of vaccine strategies in record time. RNA and adenoviral vector-based vaccines were designed, both of which induced an immunological response against the viral membrane spike protein. Within months of the introduction of these vaccines, very rare cases of thrombotic events associated with the occurrence of thrombocytopenia, mainly following adenoviral vector vaccination (AstraZeneca-Oxford and Johnson & Johnson vaccines), were reported. Briefly, three independent international research groups described this new syndrome called “vaccine-induced immune thrombotic thrombocytopenia” (VITT) [122,123,124]. The description combines several criteria, including the occurrence of a thrombotic event (usually cerebral venous thrombosis and hepatic thrombosis) within 5 to 30 days post-vaccination, and several biological criteria, including thrombocytopenia (<150 G/L), a major increase in D-dimers (>4000 ng/L), and the presence of specific platelet anti-PF4 antibodies.

While the pathophysiology is still under characterization, it has been established that VITT resembles heparin-induced thrombocytopenia (HIT) but with the formation of vaccine component-bound anti-PF4 antibodies recognized by the platelet FcγRIIa receptor, contributing to their activation and systemic thrombosis formation [122]. Post-mortem analysis of the thrombus formed indicated that they were rich in platelets and immune cells, as well as NETs, indicating the occurrence of a potent immunothrombotic reaction [125,126].

## 5. Relevance of Platelet-Derived Extracellular Vesicles

Platelet-derived extracellular vesicles (EVs) (microvesicles or microparticles) are produced upon thrombocyte activation and may be associated with infectious diseases [127]. Two main types of EVs have been described: microparticles or microvesicles (100 to 1000 nm) and exosomes (40 to 100 nm) [128]. Megakaryocytes and platelets are sources of EV production. Flow cytometry allows them to be better characterized. Indeed, EVs express CD62P and phosphatidylserine (PS), in addition to CD41 [129], unlike MK EVs, which do not express CD62P or PS but contain FilaminA [130].

Plasma from septic mice or patients is rich in EVs compared with that from healthy donors [131,132]. Some populations of platelet-derived EVs are produced to a greater extent during sepsis. Platelet-derived EVs are the most represented in the blood of septic patients in comparison with other cell-derived EVs [133,134]. EV functions in sepsis are associated with bacterial clearance [135,136]. Platelet-derived EVs propagate the proinflammatory and procoagulant responses of sepsis [56]. They carry lipid mediators [137,138] and DAMPS such as HGMB1 [139] but also cytokines [140]. Platelet-derived EVs can exert a procoagulant response to sepsis [61]. After platelet activation and the release of EVs, they can induce leukocyte aggregation [141].

Platelet-derived EVs enter the bone marrow and interact with progenitors of the megakaryocyte lineage with the reprogramming of secondary MKs [142]. Recently, it was shown that during an inflammatory situation, platelet-derived EVs interacted with bone marrow MKs resulting in functional reprogramming with inflammation-induced megakaryopoiesis [143]. Platelet microparticles have also been shown to have a role in megakaryocyte differentiation and platelet production independently of thrombopoietin in a liver injury model [142]. It has also been shown in a liver injury model that platelet microparticles play a role in megakaryocyte differentiation and platelet production independently of thrombopoietin through microRNA transport [144], which interacts with gene expression in the megakaryocyte; in this study, the platelet microparticles were filled with miR-1915-3p [142]. In contrast, during bacterial sepsis in humans, platelet microparticles are enriched in miR-223, which reduces ICAM1 expression by endothelial cells and has a protective effect by decreasing leukocyte recruitment [145]. In a *Candida albicans* infection model, the Johnnidis JB team found that this miR-223 altered the hematopoiesis in the marrow of the tested mice by promoting the production of granulocytic lineages and decreasing megakaryocyte–erythroid progenitors [146].

## 6. A Role for Specialized Megakaryocytes during Sepsis

### 6.1. Occurrence of a Specific MK Population in Sepsis?

In the medullary niche, megakaryocyte (MK) maturation is regulated by thrombopoietin (TPO), which is responsible for the specialization of hematopoietic stem and progenitor cells (HSPCs) to the pro-megakaryocyte through specific signaling induced by the TPO receptor MPL. HSPCs mature into the common myeloid progenitor cell (CMP), the MK-erythrocyte progenitor cell (MEP), and, finally, the lineage-committed megakaryocyte precursor (proMK) [147,148]. The JAK/STAT5 pathway downstream of MPL targets specific transcription factors, such as FLI1, GATA1, and RUNX1, in HSPCs to commit them into the differentiation pathway leading to platelet production [149].

In sepsis, the sharp increase in proinflammatory cytokine concentrations in the blood can spread to the bone marrow; the medullar MKs then follow either a classical or an emergency development pathway [150]. A circulating MK population may also respond to this emergency platelet production signal, as demonstrated in mice [151]. In humans, the impact of the various cytokines produced during the cytokine storm associated with SARS-CoV-2 on the MK lineage, in particular, their effect on activating or repressing their maturation and differentiation, has been recently reviewed [152]. For example, IL6, an abundant cytokine in COVID-19 [153], is associated with a significant increase in the proliferation of the MK lineage characterized by a greater number of immunoregulatory receptors (TLRs and interleukin receptors) on its surface, as evidenced in mice and humans [154,155]. In acute inflammatory conditions, the stimulation of IL6 and TNFα [156] receptors impact the maturation of MKs. In this case, activated transcription factors are independent of the JAK/pSTAT5 pathway and are responsible for the commitment of the lineage from HSPCs to stem-like MK-committed progenitors (SL-MKP) and then directly to proMKs. This accelerated MK maturation for platelet production, set up to counteract sepsis-induced thrombocytopenia, follows a specific pathway of maturation [157]. These MKs then migrate to the vascular niches of the bone marrow into the bloodstream and can reach the lungs and spleen [151,158,159]. The lung HSPC progenitors could also go the opposite way and recolonize the bone marrow [158].

This subpopulation of MKs has a strong immunological profile as shown by Liu’s team [160]. They noted an increase in the amount of mRNA and protein expression for some TLRs (TLR2 and TLR4) and membrane interleukin receptors (IL4R, IL2R, IL10R, and IL1R). Moreover, these MKs can produce some chemokines, such as CCL2, CCL9, CCL6, and lipocalin 2 [160]. These changes are observed both at the transcriptomic and protein levels in human MKs, as well as in mice, during E. coli infection [160]. This subset of the immune-specialized MK population, in fetal liver and embryonic [161], expresses CD14, a monocyte membrane marker [162]. Compared to MKs under TPO-mediated maturation, these immune-specialized MKs, even when their proliferation is induced, produce fewer platelets and exhibit a lesser load of ploidy [156,163]. Designated as a type 5-MK subpopulation (MK5) by Liu et al. [160] or MK5-derived-immune-stimulate cells (Mk5DICs) by Wang’s group [161], these immune-specialized MKs can be phenotyped and monitored in tissues through the expression of CD148 and CD48 membrane receptors. After LPS or IFNγ injection, as well as in E. coli-induced peritonitis, a circulating platelet population bearing CD148+/48+ appears within 12 h in mice bone marrow [160]. During *Listeria monocytogenes* infection, Mk5DICs disseminate into the bloodstream to infiltrate the spleen, liver, and lungs. In this mice model of infection, the circulating MK subpopulation produces TNFα and IL6, leading to an increase in phagocytosis caused by macrophages and neutrophils [164].

The MK5/Mk5DIC subpopulation is also found in the subendothelium lining the intramedullary vessels, where they can be activated by the increased intraluminal production of inflammatory mediators and chemokines. They can promote the transmigration and enrichment of neutrophils and monocytes from the medullary areas where they breed, forming a defense shield against pathogens through this MK satellite. This neutrophil recruitment can extend as far as emperipolesis, i.e., the internalization of one or more neutrophils into the MK5 cytoplasm. This phenomenon allows the fusion of neutrophil material into the platelets produced and, thus, enhances the immune content of these specialized MKs [150]. These MK5 subpopulations have a heightened ability to phagocytose pathogens, dismantle proteins, and present antigens through the acquisition of MHC-II, in addition to MHC-I [163,165]. In addition, membrane enrichment with CD148, CD48, and CD86 stabilizes the TCR synapse and facilitates antigen presentation by MCH-I and -II from these MKs to T cells [166,167]. Recently, Sun et al. [168] were able to study the appearance of this immune MK population in mice after LPS injections by performing a transcriptomic study of this CD41+/Cd53+/Lsp1+ megakaryocyte subpopulation. These MKs are found in the bone marrow and the lungs. They show an increased expression of genes involved in antigen presentation, with specialization in phagocytosis and lysosomal degradation [169,170]. Very recently, Valet et al. identified a sepsis-dependent population of splenic MKs characterized by a strong membrane expression of CD40L [151]. These specialized MKs generate a platelet population that is thought to be produced by IL3-dependent rather than TPO-dependent extramedullary megakaryopoiesis and to have immunomodulatory functions [151]. Finally, another study showed that TNFα treatment induced thrombocytopenia within 24 h and increased MK spleen infiltration in healthy mice [171]. Thus, the evidence for the existence of MKs with immune functions is increasingly compelling [169].

### 6.2. Differential Protein Content in Megakaryocytes and Platelets in Viral and Bacterial Sepsis

The bone marrow environment can be altered during bacterial or viral sepsis as a result of the production of various cytokines. During viral infection, an intercellular alert system is mediated by the secretion of type I interferon (IFN), which can diffuse from the bloodstream after the activation of circulating monocytes [172]. The presence of a viral pathogen in the bone marrow also induces the local production of type I IFN caused by leukocytes, plasmacytoid dendritic cells, and MKs themselves through direct TLR3 activation. In vitro, contact with dsRNA can also induce such a production [173,174], as well as intraperitoneal injections of Polyinosinic-polycytidylic acid (poly(IC) in mice [157]. Type I IFN has been shown to rapidly activate additional gene expression in the MK population [175], including interferon regulatory factor 7 (IRF7), human myxovirus resistance protein 1 (MxA), and IFITM3. IRF7 is a transcription factor that increases type I IFN synthesis from TLR signaling [176]. MxA is a cytoplasmic monomeric protein that aggregates around the virus capsid once entered [177].

Moreover, Boilard’s team studied the occurrence of a membrane protein expression, interferon-induced transmembrane 3 (IFITM3), on MKs during viral infection [170,178] providing antiviral properties to the generated platelets [179]. The role of this protein is to modify membrane cholesterol homeostasis in order to block the fusion of the virus particle with the endosome membrane, thus inhibiting the entrance of viral genetic material [180]. By keeping the virus particles in the endosomes, they are preferentially degraded by fusion with lysosomes in the cytoplasm. This is a common defense mechanism against many bone marrow-infecting viruses, such as H1N1 influenza, HIV, dengue [181], and the SARS-CoV family [181,182]. Overexpression of the IFITM3 system in the membrane of circulating platelets leads to increased viral clearance. Similarly, inactivation of the IFITM3 protein in a KO mouse model was found to lead to increased lethality due to the dengue virus [183]. Moreover, it has been reported that patients with a low level of IFITM3 protein expression had more severe forms of dengue [184] and H1N1 [185] infection and increased mortality, correlated with the inability of viral clearance by platelets [184].

In addition to changes in membrane protein expression in response to bacterial or viral stimuli, changes in platelet and MK contents have been reported and play a prominent role in thromboinflammation.

Recently, transcriptomic analyses of MKs in COVID-19 and sepsis in general have highlighted the overexpression of two S100A8/S100A9 genes encoding two proinflammatory myeloid-related proteins (MRP8/14), also called calprotectin proteins, which are found in granules [186]. They are secreted during platelet activation in injured areas and induce severe endothelial dysfunction [134]. The MRP8/14 complex targets TLR4 and receptor for advanced glycation endproducts (RAGE), activating multiple pathways of cellular inflammation in leukocytes and ECs [187,188]. A clinical trial to block calprotectin has been initiated in vivo with tasquinimod, a quinoline-3 carboxamide, which complexes with secreted MRP14 and prevents its binding to TLR4 and RAGE receptors [189]. This study has shown encouraging effects on inflammation mediated by these two MRP8/14 proteins. Targeting them could also prevent liver damage and bacterial dissemination in early sepsis [190].

Changes in protein expression in response to bacterial stimuli have been reported to play a prominent role in immunothrombosis during bacterial sepsis. Platelets interact with Gram-positive bacteria, such as those in the *Staphylococci* and *Streptococci* families. Their major membrane glycoproteins, GPIb and GPIIbIIIa, can directly or indirectly bind to pathogens. *Streptococcus sanguis* and *Staphylococcus aureus* have surface proteins that interact directly with the von Willebrand factor (vWF), and, subsequently, platelets adhere through GPIb [34]. Similarly to vWF, fibrinogen can coat *Staphylococcus aureus* [36] and cause platelet aggregation. Bacteria can also express proteins on their surface with an arginine–glycine–aspartic acid (RGD) sequence that binds to GPIIbIIIa, such as staphylococcus aureus [191], as well as the Gram-negative *Borrelia burgdorferi* [35]. Platelets will surround the bacteria in aggregates [32] and can subsequently phagocytize them [46]. Middleton et al. performed a platelet transcriptomic study in mice with cæcal ligation and puncture (CLP) sepsis and in humans with bacterial sepsis. They identified a significant increase in GPIIbIIIa at the platelet membrane both in mice and in patients. This increase was observed within 24 h after CLP in mice and was correlated with increased mortality [60]. These results suggest an adaptation of MKs to the presence of a bacterial pathogen.

## 7. Conclusions

Platelets are emerging as a central player in the mechanisms of thromboinflammation, particularly through interactions with other cells, the secretion of their granule content, and through the priming of some of their receptors (Figure 2). They could be compared to “primed” circulating sentinels with fast, efficient, and powerful reaction capabilities. However, their short lifespan suggests backup facilities. Such a mechanism is provided by MKs, which are emerging as true drivers of thromboinflammation, with platelets being their fast and strong-arm while their machinery is induced. A population of specialized MKs adapted to contribute to immunothrombosis appears as a true compositor of this symphony in which platelets orchestrate many interactions with the different actors.

Undoubtedly, many of the remaining fascinating questions in the field will soon be tackled.

## Figures and Tables

**Figure 1 cells-11-01507-f001:**
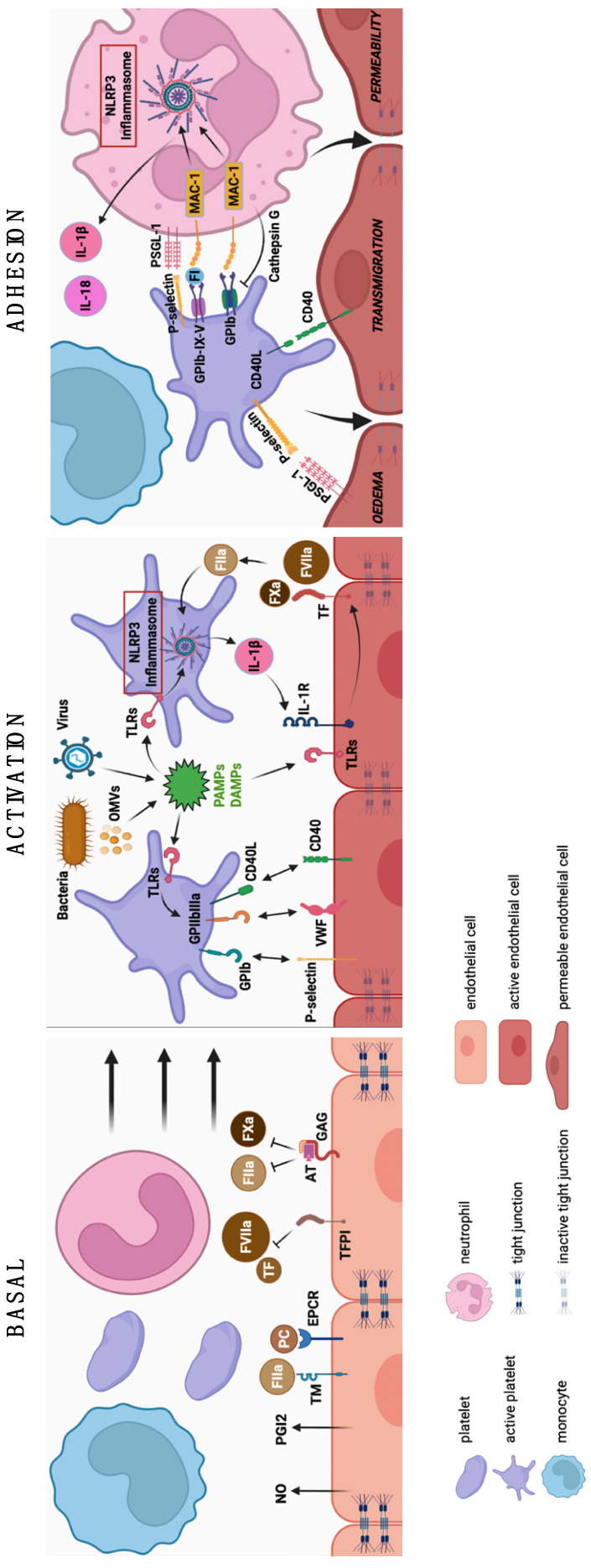
A trio of leukocyte–platelet–endothelium interactions lead to thromboinflammation associated with sepsis. Patterns of interactions between ECs, leukocytes, and platelets in the basal, activation, and adhesion steps of sepsis.

**Figure 2 cells-11-01507-f002:**
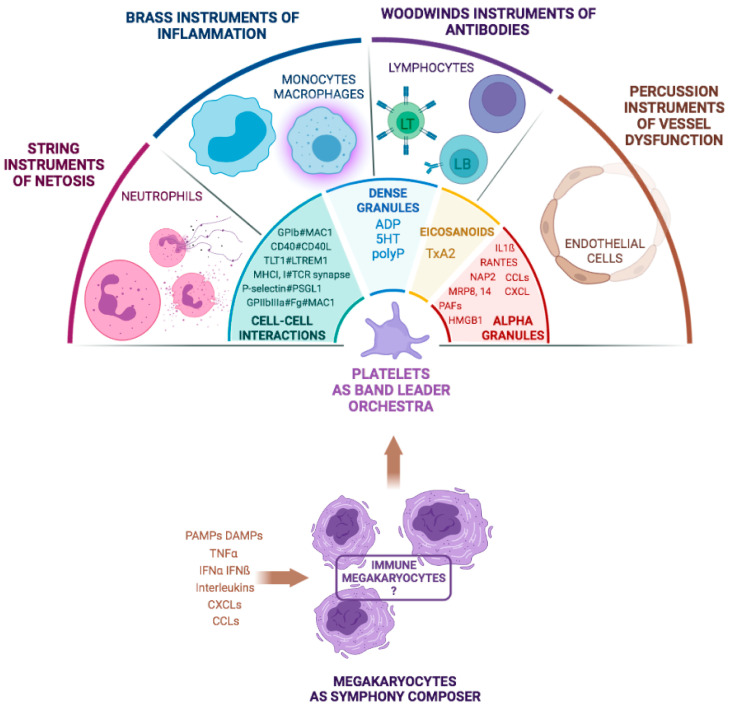
Megakaryocytes and their great orchestra in the septic scene. Platelets interact with components of the circulating blood, as well as with the endothelial cells, and are also able to guide monocytes/macrophages to subendothelial infection sites. To do this, they engage in a wide repertoire of interactions, ranging from direct cell interactions to indirect interactions via the secretion of alpha and dense granule contents and the production of eicosanoids. In this ensemble, MKs, medullary or circulating, are also able to sense and respond to certain sepsis-specific signals, such as PAMPs, DAMPs, and cytokines/chemokines. This response can be a quantitative one but also functional, i.e., a dedicated production of platelets with immune functions.

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
