# Peer review of "Platelet Versus Megakaryocyte: Who Is the Real Bandleader of Thromboinflammation in Sepsis?"

_cells, 2022, doi:10.3390/cells11091507_

Round 1
Reviewer 1 Report
The review by Garcia Cedric et al describes the role of platelets and megakaryocytes in immunothrombosis/thromboinflammation. The topics covered in this review are of general interest to the scientific community because they span multidisciplinary areas. Nevertheless, there are major concerns: Since half of the review refers to megakaryocytes, the title should include these cells. In the sections on megakaryocytes, many references are inconsistent with the text statement. Also, there is no distinction as to where the megakaryocytes come from (mice, humans, bone marrow, lung, adult, cord blood, or cell lines), giving the reader the impression that they are all the same, which is not true. The results and conclusions obtained on cell lines or mice cannot be extrapolated to human cells. Most Mk'paragraphs are related to dengue. The effects on megakaryopoiesis in other viruses or pathogens, especially bacteria- induced sepsis, should also be considered.
Missing from the review are personal opinions and reflections on the various topics covered.
Detailed concerns:
Line 33 and 35: Please write E coli first time with full name and italics and also the full name for PAMPS and DAMPs.
Line 74: Endothelial and epithelial cells exert some immune responses, but they are not considered innate immune cells. Please rephrase this sentence.
Line 76 and 77: Please delete the full spelling for PAMPS and DAMPS.
Line 83: Please spell out NFkB in full.
Line 92: Cell adhesion molecules include selectins, please delete the latter.
Line 93: Define VE-cadherin.
Line 104. define NO and PGI2.
Line 111: Delete the full notation for PAMPS and DAMPS.
Lines 110-112 ": Bacteria with their outer membrane vesicles (OMVs) and viruses produce pathogen-associated molecular patterns (PAMPs) and damage-associated molecular patterns (DAMPs) that are triggered to activate the cellular environment. This activation cascade is mediated by TLRs on platelets and ECs…" PAMPs are not produced by bacteria or viruses, they are constitutively expressed on different cell types. Some of them may be overexpressed or translocated to the membrane upon stimulation. Activation of TLRs on monocytes and polymorphonuclear leukocytes also plays an important role in activating the cellular environment, so please rephrase this sentence.
Lines 104-128. The three paragraphs should be part of the text with the appropriate references and not serve as a legend to the figure.
Many references are missing in the explanation of these paragraphs 133-135; 135-139; 149-151; 185-199. Individual references should be given rather than citing a review for all of them.
Line 212- NLRP3 should be spelled out the first time it appears in the text.
Line 221- Please replace synthesize with synthesis.
Line 231-233- LPS-induced platelet activation is quite controversial. This needs to be clarified in the paragraph with the appropriate references.
Line 239. Please clarify that P2Y12 is an adenosine diphosphate receptor.
Line 254. this paragraph is not sufficiently elaborated.
Line 294-349 I think it is not appropriate to assume that the Mks from the different studies cited in references 68-83 are the same, considering that they were performed with different Mks sources, including mice, humans, adults, cord blood, or cell lines... Please reorganize the entire paragraph 5 to clarify the origin of the mks.
Line 310. The references in this paragraph are quite old and have no relation to Covid-19. Therefore, it is important to rephrase the sentence to make it clearer that IL -6 and TNF might have an impact on megakaryopoiesis not only in Covid but also in bacteria-induced sepsis.
Line 318. Please, clarify which Mks are involved because in ref 71 they are from human bone marrow and the Mks in the study in ref 72, which has not yet appeared in Pubmed, are from cord blood and are cell lines. Also, ref 72 should be deleted because it appeared in bioRxiv in 2019 but is not yet in Pubmed.
Lines 318-32. Please, add references for this paragraph. It is not clear if you are talking about the mks studies in ref 71, 72, or both.
Line 322. The word plaquettogenic should be replaced with an English word and it is not clear why reference 74 is in this sentence.
Lines 322-325. Please, check reference 75 because this study does not show TNF-induced thrombocytopenia.
Line 376. Reference 91 does not show LPS injection in mice and mainly refers to TLT1 translocation to membrane platelets and not to megakaryocytes. The entire paragraph on TLT1 needs to be shortened.
Line 324. "These immunospecialized MKs, termed by the team of Liu et al (76) as type 5 MK subpopulation (MK5) or MK -derived-324 stimulating cells (MDICs)"... Which Mks are you referring to?
Line 327. This study was done in mice so you should not extrapolate concepts.
Lines 351-353. Please include references of interferon production by the different type of cells in the bone marrow.
Lines 357-359. Please, check references 85 and 86. I guess they should be inverted.
Lines 362-365 Please, give specific references for these items, not a general review.
The role of Mks as INF type I producing cells is very important so please, extend this paragraph to other pathogens and not just the Dengue virus.
Line 367. Please clarify that this study (ref 90) was not done specifically using platelets but whole blood and that is a transcriptomic study.
Line 372. “In addition to changes in membrane protein expression in response to bacterial or 372 viral stimuli”….This sentence is confusing as you just mentioned in the line above that IFTM3 response is specific for viruses and not bacteria…. In addition, you did not mention any other protein-membrane change due to bacterial infection….
Line 385. The study in reference 95 by Frydman GH does not use MK5 population and shows that histones are stored in alpha granules of neither mks nor platelets….
The whole paragraph of granules should be deleted.
Lines 351-353. Please, provide references to the production of interferon by other types of cells in the bone marrow.
Lines 357-359. please review references 85 and 86. I think they should be reversed.
Lines 362-365 Please provide specific references for these points, not a general review.
The role of Mks as type I interferon producing cells is especially important, so please extend this paragraph to other pathogens and not just Dengue virus.
Line 367. please clarify that this study (ref 90) was not done specifically with platelets but with whole blood and that it is a transcriptomic study.
Line 372. "In addition to changes in membrane protein expression in response to bacterial or 372 viral stimuli"....This sentence is confusing because you just mentioned in the line above that the IFTM3 response is specific to viruses and not bacteria... Also, you did not mention any other protein-membrane change due to bacterial infection...
Line 385. the study in reference 95 by Frydman GH does not use a MK5 population and shows that histones are not stored in the alpha granules of either Mks or platelets...
The entire paragraph on granules should be deleted.
Author Response
We would like to thank Reviewer 1 for all his valuable comments, which helped us to improve our work and to clarify our views.
The changes are shown in yellow in the manuscript.
Point 1: "Since half of the review refers to megakaryocytes, the title should include these cells"
Response 1: we agree with Reviewer 1 and change the title to "Platelet versus Megakaryocyte: Who is the real bandleader of thromboinflammation in sepsis?"
Point 2: "Many references are inconsistent with the text statement"
Response 2: we agree that this is the case and make the necessary changes.
Point 3: "There is no distinction as to where the megakaryocytes come from".
Response 3: we agree and many clarifications have been made in paragraph number 6, as requested.
Point 4: "Most MK'paragraph are related to Dengue".
Response 4: we agree and, globally, have indeed broadened our focus on other viruses/pathogens.
Point 5: "Please write E coli first time with full name and italics and also the full name for PAMPS and DAMPs"
Response 5: the changes can be seen on the lines 33 and 35-36 of the new text draft.
Point 6: "Endothelial and epithelial cells exert some immune responses, but they are not considered innate immune cells. Please rephrase this sentence"
Response 6: the change can be seen on the lines 80-81 of the new text draft.
Point 7: "Please delete the full spelling for PAMPS and DAMPS."
Response 7: the changes can be seen on the line 82 of the new text draft.
Point 8: "Please spell out NFkB in full."
Response 8: the changes can be seen on the line 89 of the new text draft.
Point 9: "Cell adhesion molecules include selectins, please delete the latter."
Response 9: the changes can be seen on the line 98 of the new text draft.
Point 10: "Define VE-cadherin."
Response 10: the changes can be seen on the lines 99-100 of the new text draft.
Point 11: "Define NO and PGI2".
Response 11: the change can be seen on the line 113 of the new text draft.
Point 12: "Delete the full notation for PAMPS and DAMPS".
Response 12: the change can be seen on the lines 119 and 120 of the new text draft.
Point 13: "PAMPs are not produced by bacteria or viruses, they are constitutively expressed on different cell types. Some of them may be overexpressed or translocated to the membrane upon stimulation. Activation of TLRs on monocytes and polymorphonuclear leukocytes also plays an important role in activating the cellular environment, so please rephrase this sentence."
Response 13: indeed, this sentence, which was not correct at all, has been modified, as can be seen from the lines 119 to 125 of the new text draft.
Point 14: "The three paragraphs should be part of the text with the appropriate references and not serve as a legend to the figure."
Response 14: we have preferred not to integrate the legend of Figure 1 into the body of the text to provide a more accurate set of figures and legend. However, we have added references to document our comments on the lines 113, 116, 117, 125, 128, 131, 132, 133, 134, and 140 of the new draft of the text.
Point 15: "Many references are missing in the explanation of these paragraphs 133-135; 135-139; 149-151; 185-199. Individual references should be given rather than citing a review for all of them."
Response 15: we fully agree with Reviewer 1 and have included bibliographic references, taking care to cite articles rather than reviews, as can be seen on the lines 144, 147, 148, 149, 150, 152, 154, 156, 158, 160, 163, 170, 172, 174, 176, 177, 179, 180, 184, 185, 187, 192, 198, 201, 202, 205, 211, 215, 217, and 220 of the new draft of the text.
Point 16: "NLRP3 should be spelled out the first time it appears in the text"
Response 16: the change can be seen on the line 246 of the new text draft.
Point 17:"Please replace synthesize with synthesis."
Response 17: the change can be seen on the line 255 of the new text draft.
Point 18: "Please clarify that P2Y12 is an adenosine diphosphate receptor."
Response 18: the change can be seen on the lines 273-274 of the new text draft.
Point 19: "this paragraph is not sufficiently elaborated."
Response 19: indeed, we have expanded this part of our work; the change can be seen on the lines 288 to 322 of the new text draft.
Point 20: " I think it is not appropriate to assume that the Mks from the different studies cited in references 68-83 are the same, considering that they were performed with different Mks sources, including mice, humans, adults, cord blood, or cell lines... Please reorganize the entire paragraph 5 to clarify the origin of the mks."
Response 20: we have indeed lacked precision and regret this; we have indeed rewritten the whole paragraph and included the necessary references and clarifications in order not to over-interpret the published data. We hope that the clarifications made will meet the expectations of the reviewer 1. All references associated with this section have been revised
the structure of this paragraph has also been changed and the elements on granules have been removed. The changes can be seen on the lines 379 to 536 of the new text draft.
Reviewer 2 Report
In this review, the authors analyze and discuss the ambivalent role of platelets as a host defensive system, on one hand, in interacting with pathogens, and, on the other hand, as contributors to the development of sepsis complications, particularly focusing on some key mechanisms that can lead to thromboinflammation/immunothrombosis during sepsis.
The title is attractive, the review is well organized and well written, and it not only fully illustrates the current knowledge in this field, but also indicates those aspects that require further investigation in the future.
We would suggest, in order to improve the manuscript, to address few points:
- in the paragraph “From leuko-platelets interactions to the involvement in NETosis”, the role of platelets as potent triggers of NETosis through the binding of platelet P-selectin and high mobility group protein B1 (HMGB1) to pattern recognition receptors (such as Toll-like receptor 4 (TLR4) and receptor for advanced glycation end products (RAGE)), complement receptor C3aR and P-selectin glycoprotein ligand 1 (PSGL1) on neutrophils could be analyzed in more detail.
- The paragraph “Platelet and endothelial cells (ECs) crosstalk during sepsis” could be expanded and include some recently published studies.
- Also the paragraph “Effects of antiplatelet drugs in sepsis” could be expanded including other recent studies.
- Moreover, recent studies underline the relevance of platelet-derived extracellular vesicles in influencing platelet behavior and their role in several critical aspects of sepsis and disseminated intravascular coagulation. We believe that adding a paragraph on the role of extracellular vesicles in sepsis may ameliorate this review by increasing its completeness.
Please also note that bibliographic references are missing in several places in the text.
Finally, also check the following minor points:
- Page 1, line 30, check the spacing.
- Page 4, line 121-125, two sentences are repeated .
- Page 4, line 131, please check the grammar.
- Page 7, line 265, check the SARS-CoV-2 spacing.
Author Response
First of all, we would like to thank Reviewer 2 for all his feedback which led us to modify our manuscript and upgrade its content.
The modifications related to his comments are highlighted in green in the manuscript.
Point 1: "in the paragraph “From leuko-platelets interactions to the involvement in NETosis”, the role of platelets as potent triggers of NETosis through the binding of platelet P-selectin and high mobility group protein B1 (HMGB1) to pattern recognition receptors (such as Toll-like receptor 4 (TLR4) and receptor for advanced glycation end products (RAGE)), complement receptor C3aR and P-selectin glycoprotein ligand 1 (PSGL1) on neutrophils could be analyzed in more detail."
Response 1: we have not completely rewritten this paragraph, but we have added the sentence lines 184 to 187 on the Complement that we were not referring to. Finally, we have also attached a number of references that were missing. The changes are shown on lines 144, 147, 148, 149, 150, 152, 154, 156, 158, 160, 163, 170, 172, 174, 176, 177, 179, 180, 184, 185, 187, and 192 of the new manuscript.
Point 2: "The paragraph “Platelet and endothelial cells (ECs) crosstalk during sepsis” could be expanded and include some recently published studies."
Response 2: Again, an additional effort has been made on the bibliographic references, which were poor. The changes are indicated in lines 198, 199, 201, 202, 205, 207, 2011215, 217, and 220 of the new manuscript. The section on TLT-1 has been inserted in this paragraph to illustrate the interactions between endothelial cells and platelets, as can be seen from lines 225 to 241 of the new manuscript.
Point 3: "Also the paragraph “Effects of antiplatelet drugs in sepsis” could be expanded including other recent studies."
Response 3: the paragraph on the effects of antiplatelet therapy has also been strengthened by data from meta-analyses on aspirin in particular, work on P2Y12 inhibitors, P-selectin, CLEC-2. The changes were also requested by Reviewer 1 and can be seen from lines 288 to 322 of the new manuscript.
Point 4: "Moreover, recent studies underline the relevance of platelet-derived extracellular vesicles in influencing platelet behavior and their role in several critical aspects of sepsis and disseminated intravascular coagulation. We believe that adding a paragraph on the role of extracellular vesicles in sepsis may ameliorate this review by increasing its completeness."
Response 4: we fully agree with Reviewer 2 and have included paragraph 5 in the new microparticle manuscript which is visible from lines 345-377.
Point 5: "line 30, check the spacing."
Response 5: this element has been corrected as can be seen in line 30 of the new manuscript.
Point 6: "line 121-125, two sentences are repeated."
Response 6: this element has been corrected as can be seen in the lines 134 to 137 of the new manuscript.
Point 7: "line 131, please check the grammar."
Response 7: this element has been corrected as can be seen in the lines 142 to 144 of the new manuscript.
Point 8: "line 265, check the SARS-CoV-2 spacing. "
Response 8: this element has been corrected as can be seen in the lines 306 of the new manuscript.
Reviewer 3 Report
In this article, Garcia et. al. is highlighting the accumulated evidence regarding the emerging roles and functions of blood platelets in innate immunity and inflammatory response in sepsis. The authors also discussed the specialized an immune - a disease specific - subpopulation of megakaryocytes and their generated platelets that have a immunomodulatory roles in pathogen infection.
The following issues should be addressed/further discussed by the authors:
- Despite their short lifespan, proinflammatory reprogramming of platelets might be an important route in the onset of the species infection!
- The major role of medullary and circulating megakaryocytes in sepsis.
- The disease severity and thrombocytopenia during the septic shock.
- Platelets/Megakaryocytes dynamics as determinants of clinical outcomes in sepsis patients.
- In the introduction line 38, the authors mentioned that they are going to present the ambivalent role of platelet in sepsis, so it will be great if the author can highlight the contradictory roles that he was able to find the literature in a table in a way that can be compared easily the good vs the bad! that will add value to the manuscript.
- There seems to be a typo in section two immunothrombosis or thromboinflammation. In line 48-49 "subtill" perhaps the author meant "subtle"
- Editing of the sentence (Line 23) for clarity.
- The title might be edited for clarity (line 271).
Author Response
First of all, we would like to thank Reviewer 3 for all his comments which led us to modify our manuscript and improve its content. Some of the suggested improvements, although relevant and in particular, on thrombocytopenia, have not been systematically followed due to the limited number of words already largely exceeded. However, the paragraph on megakaryocytes has been extensively revised and should meet the requested clarification efforts. Similarly, Table 1 listing some of the positive and negative effects of platelets in sepsis should provide a synthetic view of this ambivalent role of platelets.
Changes related to his comments are highlighted in pink in the manuscript.
Point 1: "In the introduction line 38, the authors mentioned that they are going to present the ambivalent role of platelet in sepsis, so it will be great if the author can highlight the contradictory roles that he was able to find the literature in a table in a way that can be compared easily the good vs the bad! that will add value to the manuscript. "
Response 1: A table listing some of the beneficial and deleterious elements of platelets has been inserted between the lines 71 and 72 of the new manuscript, as suggested by Reviewer 3.
Point 2: "There seems to be a typo in section two immunothrombosis or thromboinflammation. In line 48-49 "subtill" perhaps the author meant "subtle".
Response 2: the word has been changed line 52 of the new manuscript.
Point 3: "Editing of the sentence (Line 23) for clarity."
Response 3: the sentence has been changed line 23 of the new manuscript.
Point 4: "The title might be edited for clarity (line 271)."
Response 4: the sentence has been changed line 324 and 325 of the new manuscript.
Round 2
Reviewer 1 Report
The review has been significantly improved. There are two minor comments that the authors should still address. GPIB in lines 491 and 494 should be replaced by GPIb.
IFN1 in lines 448 453 and 450 should be replaced by type I IFN and should be spelled the first time it appears and not later as it is in line 456.
Author Response
We are grateful to have had the benefit of Reviewer 1's comments, which allowed us to improve our message and to make the clarifications that were missing.
Point 1: "GPIB in lines 491 and 494 should be replaced by GPI."
Response 1: The 2 changes, lines 491 and 494, have been made and highlighted in yellow in the text.
Point 2: "IFN1 in lines 448 453 and 450 should be replaced by type I IFN and should be spelled the first time it appears and not later as it is in line 456."
Response 2: Type I IFN was spelled correctly on line 448 and changed on lines 450, 453 and 456, as highlighted in yellow in the text.
Reviewer 3 Report
We are satisfied with the authors response.
Author Response
We are grateful for Reviewer 3's comments, which helped us improve our message.